# Inner Retinal Layer Changes Reflect Changes in Ambulation Score in Patients with Primary Progressive Multiple Sclerosis

**DOI:** 10.3390/ijms241612872

**Published:** 2023-08-17

**Authors:** Jonathan A. Gernert, Luise Böhm, Michaela Starck, Stefan Buchka, Tania Kümpfel, Ingo Kleiter, Joachim Havla

**Affiliations:** 1Institute of Clinical Neuroimmunology, Ludwig-Maximilians-Universität München, 81377 Munich, Germany; 2Marianne-Strauß-Klinik, Behandlungszentrum Kempfenhausen für Multiple Sklerose Kranke gGmbH, 82335 Berg, Germany; 3Institute of Medical Information Processing, Biometry, and Epidemiology, Faculty of Medicine, Ludwig-Maximilians-Universität München, 81377 Munich, Germany; 4Biomedical Center and University Hospital, Ludwig-Maximilians-Universität München, 82152 Planegg-Martinsried, Germany; 5Department of Neurology, Ruhr-University Bochum, 44791 Bochum, Germany

**Keywords:** multiple sclerosis (MS), relapsing–remitting multiple sclerosis (RRMS), primary progressive multiple sclerosis (PPMS), optical coherence tomography (OCT), ambulatory functional score

## Abstract

The establishment of surrogate markers to detect disability progression in persons with multiple sclerosis (PwMS) is important to improve monitoring of clinical deterioration. Optical coherence tomography (OCT) could be such a tool. However, sufficient longitudinal data of retinal neuroaxonal degeneration as a marker of disease progression exist only for PwMS with a relapsing–remitting course (RRMS) so far. In contrast, longitudinal data of retinal layers in patients with primary-progressive MS (PPMS) are inconsistent, and the association of OCT parameters with ambulatory performance in PwMS has rarely been investigated. We aimed to investigate the relative annual rates of change in retinal layers in PwMS (RRMS and PPMS) compared with healthy controls (HC) using OCT and to evaluate their association with ambulatoryfunctionalscore (AS) worsening in PPMS. A retrospective analysis of a longitudinal OCT dataset of the retinal layers of PwMS and HC from two MS centers in Germany was performed. Walking ability was measured over a standardized distance of 500 m, and changes during the observation period were categorized using the AS and the expanded disability status scale (EDSS). 61 HC with 121 eyes and 119 PwMS (PPMS: 57 patients with 108 eyes; RRMS: 62 patients with 114 eyes) were included. The median follow-up time for PwMS was 3 years. The relative annual change of pRNFL (peripapillary retinal nerve fiber layer) and INL (inner nuclear layer) was significantly different in PwMS compared with HC. RRMS and PPMS subgroups did not differ in the annual atrophy rates. In patients with PPMS, worsening of the AS was significantly associated with increased thinning of the TMV (total macular volume), GCIP (ganglion cell and inner plexiform layer), and ONPL (outer nuclear and outer plexiform layer) (all *p*-value < 0.05, *r* > 0.30). For every −0.1% decrease in the TMV, GCIP, and ONPL, the risk of a deterioration in the AS increased by 31% (hazard ratio (HR): 1.309), 11% (HR: 1.112), and 16% (HR: 1.161), respectively. In addition, worsening EDSS in PPMS was significantly associated with the relative annual atrophy rates of pRNFL, TMV, and GCIP (all *p*-value < 0.05). Disability progression in PPMS can be measured using OCT, and increasing annual atrophy rates of the inner retinal layers are associated with worsening ambulation. OCT is a robust and side-effect-free imaging tool, making it suitable for routine monitoring of PwMS.

## 1. Introduction

Disability progression in PwMS is predominantly documented as a worsening in walking ability (ambulatory functional score, AS) with an increase in EDSS (expanded disability status scale) score. However, the interval walking distance score is highly dependent on daily performance as well as other internal factors [1,2,3]. Therefore, it is necessary to use alternative, reliable, reproducible, and validated surrogate markers to determine disability progression independent of the deterioration of the AS as part of the EDSS, the current gold standard.

Optical coherence tomography (OCT) could be such a marker: OCT has been used to study retinal imaging markers in persons with multiple sclerosis (PwMS) [4]. However, most analyses were performed in cohorts with relapsing–remitting MS (RRMS), and results about primary-progressive MS (PPMS) are limited and so far inconsistent: No significant changes in retinal layer thicknesses and/or volumes were reported in longitudinal studies for PPMS [5,6]. One study demonstrated that subjects with PPMS show less retinal atrophy compared with patients with other disease courses of MS (RRMS and secondary-progressive MS (SPMS)) [7]. In contrast, increased retinal thinning has been described in PPMS compared with healthy controls (HC), RRMS, and/or SPMS cohorts [8,9,10,11,12,13]. In addition, retinal layer thickness in PwMS has mostly been studied in relation to the EDSS score, visual acuity, or cognitive function [14,15,16,17].

To demonstrate that OCT may be a surrogate marker of disability progression in PPMS, we performed a retrospective cross-sectional and longitudinal cohort analysis of PwMS compared with HC and examined the association with worsening AS in PPMS: First, we compared OCT parameters cross-sectionally. Second, the relative annual change rates of different retinal layers were analyzed between the cohorts. Third, an association between longitudinal OCT parameters and walking ability was evaluated in PPMS.

## 2. Results

### 2.1. Study Cohorts

In total, 180 subjects and 343 eyes were included: 61 HC with 121 eyes and 119 PwMS with 222 eyes (PPMS: 57 patients with 108 eyes; RRMS: 62 patients with 114 eyes). The three cohorts (HC, RRMS, and PPMS) differed significantly in terms of gender distribution, age at baseline OCT (OCT_bas_), number of follow-up OCT examinations included, and follow-up duration in months. PPMS patients were more often male, were older (mean ± standard deviation (SD) in years: 45.1 ± 10.1), had a higher EDSS score at OCT_bas_ (median: 4.3), and had a longer disease duration (initial manifestation to OCT_bas_) (median: 7 years) than the RRMS cohort (age: 30.5 ± 10.5; *p* < 0.001; EDSS score: 2.0; *p* < 0.001; disease duration: 2 years; *p* < 0.002). The median overall follow-up time for PwMS was 3 years (interquartile range (IQR): 2–5). Demographic and clinical data are reported in Table 1.

### 2.2. Cross-Sectional Retinal Layer Analysis

In a gender- and age-adjusted mixed linear model, PwMS showed a significantly reduced pRNFL thickness compared with HC at OCT_bas_ (mean in µm: 94.5 vs. 100.2; *p* = 0.002). Additionally, the TMV and GCIP volumes were significantly decreased in PwMS in comparison to HC (both *p* < 0.001). When comparing RRMS and PPMS patients in terms of retinal layers at OCT_bas_, no significant differences were found after adjustment for *gender*, *age at OCT_bas_*, and *disease duration*. However, the mean absolute pRNFL thickness was higher in PPMS than in RRMS, with a subthreshold effect (mean in µm: 96.5 vs. 91.5; *p* = 0.072). The cross-sectional retinal layer analysis at OCT_bas_ is reported in Table 2.

### 2.3. Longitudinal Retinal Layer Analysis

For the longitudinal analysis of retinal layers, relative annual change rates were calculated (in %) (Table 3). A significantly higher annual decrease in the pRNFL thickness was detected in PwMS compared with HC (mean in %: −0.73 vs. −0.13; *p* = 0.002). The mean annual INL volume change significantly differed between HC and PwMS (mean in %: 1.21 vs. 0.33; *p* = 0.005), an additional age-related analysis for patients with RRMS or PPMS is provided in Appendix A. The comparison between patients with RRMS and PPMS disease courses showed no significant differences in the relative annual change rates. However, the mean decrease in the GCIP volume was higher in RRMS vs. PPMS (mean in %: −0.60 vs. −0.16; *p* = 0.077).

### 2.4. Deterioration in Ambulatory Function Associates with Macular Layer Degeneration

Data on AS measured in routine clinical care were available from 87 PwMS for GpRNFL (PPMS *n* = 37; RRMS *n* = 50) with a median observation time of 37 months (IQR: 23.5–61) and 48 PwMS for macular layers (PPMS *n* = 17; RRMS *n* = 31) with a median observation time of 27 months (IQR: 18.75–42.5). A total of 16 of 37 patients with PPMS deteriorated (Table 1). A comparison of OCT parameters between RRMS and PPMS patients with stable AS during the observation period revealed no differences, apart from the relative atrophy rate of the GCIP, which was significantly increased in subjects with RRMS (*p*-value = 0.004) (Table 4). The eyes of the PPMS subjects with a deterioration in the AS (during OCT_bas_ to OCT_lfu_) showed significantly higher mean relative annual decrease rates in the TMV, GCIP, and ONPL compared with eyes of PPMS subjects with stable AS (*t*-test: all *p*-value < 0.05; ANCOVA: all *p*-value < 0.05; all *r* > 0.3) (Table 4). For every −0.1% decrease in the TMV, GCIP, and ONPL, the risk of a deterioration in the AS increased by 31% (hazard ratio (HR): 1.309), 11% (HR: 1.112), and 16% (HR: 1.161), respectively, in PPMS (Table 5). Moreover, a deterioration in the EDSS in PPMS individuals was significantly associated with the relative annual atrophy rates in the pRNFL, TMV, and GCIP (all *p*-value < 0.05) (Table 5).

## 3. Discussion

Disease progression in PwMS is currently assessed primarily based on medical history and by means of clinical tests, in particular the EDSS and the AS. However, clinical testing can be subject to variability due to external and individual causes, making it difficult to assess disease progression as an individual follow-up parameter. Our aim was to investigate OCT as a surrogate to assess MS progression. In patients with PPMS, we found an association of longitudinal retinal neuroaxonal layer-changes with walking capacity as assessed by the AS: PPMS individuals with AS worsening over the median observation period of 27 months had an increased retinal neuroaxonal volume loss of the TMV, GCIP, and ONPL, compared with PPMS persons with stable AS. Higher retinal atrophy rates were also associated with a deterioration in the EDSS. To the best of our knowledge, the association between walking ability and retinal neuroaxonal layer measurements has so far only been investigated cross-sectionally [18]. In that study, a significant association between slowed walking speed (measured with the timed 25-foot walk test, T25FW) and reduced TMV, but not pRNFL, was shown. This indicates, together with our results, that only the atrophy rates of the macular layers represent additional markers of disease progression. We emphasize that the interval censoring poses a limitation of our Cox-regression model.

In addition, we report cross-sectional and longitudinal analyses of the retinal layers in RRMS, PPMS, and HC: In our cross-sectional approach, we found a significantly reduced pRNFL thickness as well as reduced TMV and GCIP volumes in PwMS compared with HC. Retinal neuroaxonal degeneration is a well-studied finding in eyes of PwMS with or without optic neuritis [19,20]. However, we detected no significant differences between PPMS and RRMS in the cross-sectional analysis of retinal layers adjusted for age, gender, and disease duration. In our longitudinal analysis, using age and gender as covariates, we found an overall increased thinning of the pRFNL in PwMS vs. HC. Similar results have been repeatedly reported in previous studies [13,21]. Additionally, we detected a significant difference in the relative annual change rate of INL volume between PwMS and HC. We emphasize that several authors reported influencing factors on the INL analysis, including physiological variations [22], clinical MS activity during the observation period [23], and immunotherapy effects [24]. In addition, there is an ongoing debate about the INL change rate in relation to age and MS disease duration [13,25,26,27]. In our cohort, we observed swelling of the INL in younger PwMS, whereas INL thinning occurred with age. In summary, our results support the hypothesis of Cordano et al., interpreting the change rate of INL in PwMS as *early inflammation followed by later neurodegeneration* [25]. As caveats, we state that we report on INL volumes, while INL thickness rates of change have been reported in the cited literature [13,25]. Further, our analyzed cohorts differ significantly in demographic and clinical parameters. Within our analyzed cohorts, we did not detect any significant differences in the rates of change of the retinal layers between PPMS and RRMS.

The main result of our investigation is the association between gait deterioration and increased longitudinal atrophy rates of retinal layers in PPMS. OCT might therefore provide imaging markers to assess disease progression in PwMS in the future. Further, we would like to emphasize the long duration of observation, especially of the progressive MS subjects in our cohort. Limitations result from the retrospective approach and the resulting restrictions on data availability, the different devices used for OCT examinations, and the reduced sample size of PwMS with available data on ambulatory performances (AS). Also, based on the retrospective analysis, it cannot be excluded that subclinical comorbidities also have an influence on the measured neuroaxonal degeneration of the retina, which underlines the explorative approach of this analysis. A prospective, multicenter study with a structured homogenized clinical dataset is needed for further analysis between walking ability and macular degeneration in PwMS.

## 4. Materials and Methods

### 4.1. Study Design

For the analysis, we retrospectively created a joint dataset from two large MS centers in southern Germany: (i) Institute of Clinical Neuroimmunology, Ludwig-Maximilians-Universität München, Munich (tertiary center; LMU Hospital) and (ii) Marianne-Strauß-Klinik, Treatment Center Kempfenhausen for Patients with Multiple Sclerosis, Berg (secondary center; MSK). Inclusion criteria were (i) all PwMS (RRMS and PPMS) with the presence of at least two OCT examinations with an interval of at least six months; (ii) no history of or occurrence of optic neuritis in the observation period; (iii) age >18 years at OCT_bas_. OCT examinations were performed from 2011 to 2022 as part of routine clinical care. Diagnoses of RRMS and PPMS were made according to the revised McDonald criteria 2017 [28]. Exclusion criteria were (i) eyes with anamnestic history of optic neuritis; (ii) eyes with a refraction error > 5 diopters; (iii) patients with a history of a known disease affecting the visual system. If available, we considered the maximum walking distance in PwMS (up to 500 m) on the dates of OCT_bas_ and the last follow-up OCT examination (OCT_lfu_), using the AS (Neurostatus Version 04/10.2 modified from [29,30]) in a binary manner (improved/stable vs. deterioration). In PPMS subjects with available AS data, the EDSS was also evaluated accordingly. RRMS subjects with an AS deterioration possibly due to, or as part of a relapse during the observation period (*n* = 9 patients), were excluded from further analysis to evaluate the association of OCT parameters and relapse-independent progression in ambulatory function. Next to the mentioned inclusion/exclusion criteria, HC had no diagnosed neurological or ophthalmological disease. All individuals examined at LMU gave written consent, and the retrospective data of the individuals included from MSK were irreversibly anonymized. The local ethic committee gave approval for a retrospective analysis of OCT datasets (project number: 19-570). This study was conducted according to the Declaration of Helsinki.

### 4.2. Optical Coherence Tomography (OCT)

Spectral domain OCT examinations were performed using a Spectralis SD-OCT at LMU (Heidelberg Engineering, Heidelberg, Germany, OCT2-Module, automated retinal layer segmentation by Heyex v2.5.5, Heidelberg Engineering, Heidelberg, Germany) and a Cirrus HD-OCT 4000 at MSK (Carl Zeiss Meditec, Jena, Germany, version 8.1.0.117). A 3 mm ring scan centered on the optic nerve head was performed to assess the peripapillary retinal nerve fiber layer (pRNFL) thickness (in µm). The pRNFL thickness was analyzed for cohorts from both centers after including a conversion factor [31]. In addition, the volumes (mm^3^) of four retinal layers acquired from a macula scan (25 [30 × 25°, ART 13] vertical b-scans) were evaluated only for subjects examined at LMU (6 mm ring centered on the fovea, not available for the MSK): (i) total macular volume (TMV); (ii) combined ganglion cell and inner plexiform layer (GCIP), (iii) inner nuclear layer (INL); (iv) combined outer plexiform and outer nuclear layer (ONPL) (Figure 1). As far as possible considering the listed inclusion and exclusion criteria, both eyes of one person were analyzed.

The different layers are shown in a horizontal OCT scan centered on the middle of the fovea. The layer segmentation is done automatically by the segmentation tool of Heyex v2.5.5. Analyzed retinal structures of this work were the total macular volume (TMV); combined ganglion cell and inner plexiform layer (GCIP), inner nuclear layer (INL); and combined outer nuclear and outer plexiform layer (ONPL). Additionally, the macular retinal nerve fiber layer (mRNFL) is shown. However, in the present analysis, the peripapillary RNFL (pRNFL) determined with a 3 mm ring scan centered on the optic disc was evaluated (in µm).

### 4.3. Statistical Analysis

SPSS Statistics 27 (IBM, Armonk, New York, United States of America) was used for statistical analysis. For comparison of nominally distributed features the Chi-squared-test was used, for metric data the *t*-test was applied, if normally distributed, otherwise the Kruskal-Wallis-test. A generalized mixed linear model was performed to evaluate the OCT parameters cross-sectionally (at OCT_bas_) and longitudinally (relative annual change rates) between cohorts: (i) HC vs. PwMS: Here, *gender* and *age at OCT_bas_* were considered as covariates; (ii) PPMS vs. RRMS: Additionally, the *disease duration* (years from initial manifestation to OCT_bas_) was considered. By including a random effect for the *centers*, we could not determine a high variance between the centers. Therefore, *center* was not included as a random variable. For the analysis of relative annual rates of change in retinal layers, extreme outliers were excluded according to Tukey (1st quartile – 3xIQR < range included < 3rd quartile + 3xIQR; interquartile range (IQR)) [32]. A *t*-test and an ANCOVA analysis (covariates *gender* and *age at OCT_bas_*) were used to assess associations between AS and change rates of retinal layers. Further, a Cox-regression (covariates *gender* and *age at OCT_bas_*) was performed to evaluate the effect of relative annual atrophy rates of retinal layers (*relative annual rates of change in retinal layers*x − 1; in ‰) on deterioration in the AS in PwMS. In PPMS subjects, also a Cox-regression (covariates *gender* and *age at OCT_bas_*) was performed to evaluate the effect of relative annual atrophy rates of retinal layers (*relative annual rates of change in retinal layers*x − 1; in ‰) on deterioration in the EDSS. Significance was set at *p* < 0.05.

## Figures and Tables

**Figure 1 ijms-24-12872-f001:**
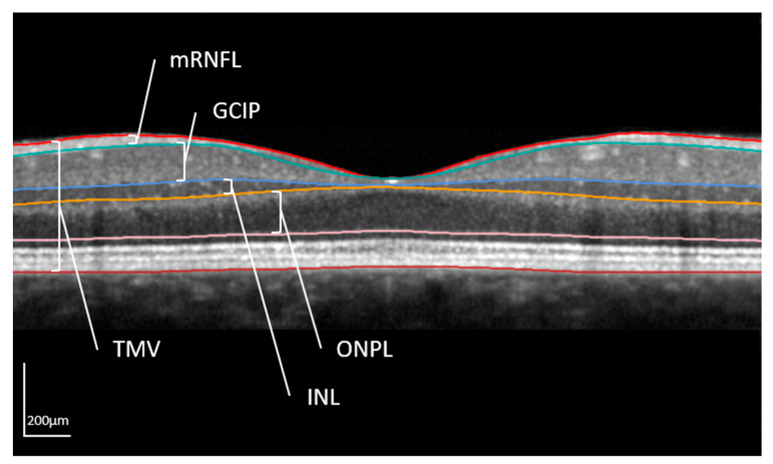
Macula scan in optical coherence tomography.

**Table 1 ijms-24-12872-t001:** Demographic and clinical data.

	HC	PPMS	RRMS	*p*-Value HC vs. PPMS	*p*-Value HC vs. RRMS	*p*-Value PPMS vs. RRMS
General Demographic Data						
Total subjects (*n*)	61	57	62			
Total eyes (*n*) Eyes scanned at Cirrus (*n*) Eyes scanned at Spectralis (*n*)	121 0 121	108 44 64	114 39 75			
Female, *n* (% of subjects)	38 (62%)	25 (44%)	40 (65%)	***0.045*** ^1^	*0.798* ^1^	***0.024*** ^1^
Age at OCT_bas_, mean ± SD (years)	36.64 ± 12.82	54.12 ± 10.68	37.19 ± 10.75	***<0.001*** ^2^	*0.796* ^2^	***<0.001*** ^2^
Number of OCT scans, median (range: minimum to maximum)	2 (2–5)	3 (2–7)	3 (2–7)	***<0.001*** ^3^	***<0.001*** ^3^	*0.377* ^3^
Follow-up time, median (IQR) (months)	7 (6–21)	40 (22–76.2)	28 (18.75–64.5)	***<0.001*** ^3^	***<0.001*** ^3^	*0.060* ^3^
Multiple-Sclerosis-Specific Data						
Age at initial onset, mean ± SD (years)		45.09 ± 10.10	30.52 ± 10.53			***<0.001*** ^2^
Disease duration, median (IQR) (years)		7 (4–10.5)	2 (0–10)			***0.002*** ^3^
Immunotherapy during observation period, *n* (treated subjects)/N (information available) (% of N)		34/44 (77%)	53/59 (90%)			*0.082* ^1^
EDSS at OCT_bas_, median (IQR)		4.25 (3.00–6.00)	2.0 (1.75–3.50)			***<0.001*** ^2^
Deterioration in ambulatory functional score (AS), *n* (deteriorated subjects)/N (information available) (% of N)		16/37 (43%)	9/50 (18%)			***0.010*** ^1^

In total, 180 subjects and 343 eyes were included. Comparison of demographic and multiple-sclerosis-specific data are shown at baseline optical coherence tomography (OCT_bas_). Significance was set at *p* < 0.05. EDSS: expanded disability status scale; HC: healthy controls; IQR: interquartile range; *n*: number; OCT: optical coherence tomography; PPMS: primary-progressive multiple sclerosis; RRMS: relapsing–remitting multiple sclerosis; SD: standard deviation. ^1^ Chi-squared test. ^2^ *t*-Test. ^3^ Kruskal–Wallis test.

**Table 2 ijms-24-12872-t002:** Cross-sectional retinal layer analysis at baseline OCT.

	HC vs. PwMS				PPMS vs. RRMS			
	HC	PwMS	*ß* (SE)	*p*-Value ^1^	PPMS	RRMS	*ß* (SE)	*p*-Value ^2^
Peripapillary Scan (MSK and LMU)								
pRNFL, mean in µm (95% CI) N (eyes)	**100.24** **(97.36–103.12)** 121	**94.47** **(92.43–96.52)** 222	** *−5.765* ** ** *(1.82)* **	** *0.002* **	96.54 (92.99–100.10) 108	91.45 (87.99–94.91) 114	*5.091* *(2.81)*	*0.072*
Macula Scan (LMU)								
TMV, mean in mm^3^ (95% CI) N (eyes)	**8.85** **(8.76–8.94)** 118	**8.62** **(8.54–8.71)** 138	** *−0.221* ** ** *(0.06)* **	** *<0.001* **	8.66 (8.54–8.79) 63	8.55 (8.43–8.66) 75	*0.115* *(0.10)*	*0.232*
GCIP, mean in mm^3^ (95% CI) N (eyes)	**1.14** **(1.11–1.16)** 118	**1.05** **(1.02–1.07)** 138	** *−0.093* ** ** *(0.01)* **	** *<0.001* **	1.05 (1.02–1.09) 63	1.02 (0.99–1.06) 75	*0.033* *(0.03)*	*0.266*
INL, mean in mm^3^ (95% CI) N (eyes)	0.97 (0.95–0.98) 118	0.98 (0.96–0.99) 138	*0.008* *(0.01)*	*0.434*	0.97 (0.95–1.00) 63	0.97 (0.95–0.99) 75	*0.003* *(0.02)*	*0.850*
ONPL, mean in mm^3^ (95% CI) N (eyes)	2.593 (2.529–2.657) 120	2.614 (2.555–2.674) 130	*0.022* *(0.05)*	*0.630*	2.632 (2.563–2.700) 63	2.571 (2.505–2.638) 67	*0.060* *(0.05)*	*0.269*

Retinal layer thickness and volumes at baseline OCT (OCT_bas_) were compared among HC and PwMS, respectively, in PPMS and RRMS. Significance was set at *p* < 0.05. *ß*: beta regression; CI: confidence interval; GCIP: ganglion cell and inner plexiform layer; HC: healthy controls; INL: inner nuclear layer; LMU: Ludwig-Maximilians Universität München (Spectralis OCT); MSK: Marianne-Strauß-Klinik (Cirrus OCT); N (eyes): number of eyes included; ONPL: outer nuclear and outer plexiform layer; PPMS: primary-progressive multiple sclerosis; pRNFL: peripapillary retinal nerve fiber layer; PwMS: persons with multiple sclerosis (PPMS and RRMS); RRMS: relapsing–remitting multiple sclerosis; SD: standard deviation; SE: standard error; TMV: total macular volume. ^1^ Generalized linear mixed model using *gender* and *age at OCT_bas_* as covariates to compare HC vs. PwMS. ^2^ Generalized linear mixed model using *gender*, *age at OCT_bas_*, and *disease duration* (initial manifestation to OCT_bas_ in years) to compare MS subtypes.

**Table 3 ijms-24-12872-t003:** Relative annual change rates of retinal layers.

	HC vs. PwMS				PPMS vs. RRMS			
	HC	PwMS	*ß* (SE)	*p*-Value ^1^	PPMS	RRMS	*ß* (SE)	*p*-Value ^2^
Peripapillary Scan, relative annual change (MSK and LMU)								
pRNFL mean in % (95% CI) N (eyes)	**−0.131** **(−0.436–0.174)** 102	**−0.730** **(−0.942–−0.519)** 208	** *−0.600* ** ** *(0.191)* **	** *0.002* **	−0.597 (−0.877–−0.317) 100	−0.833 (−1.103–−0.562) 108	*0.235* *(0.218)*	*0.282*
Macula Scan, relative annual change (LMU)								
TMV mean in % (95% CI) N (eyes)	−0.076 (−0.234–0.083) 110	−0.188 (−0.332–−0.044) 133	*−0.112* *(0.110)*	*0.309*	−0.226 (−0.417–−0.036) 61	−0.200 (−0.371–−0.028) 72	*−0.027* *(0.145)*	*0.854*
GCIP mean in % (95% CI) N (eyes)	−0.462 (−0.714–−0.209) 112	−0.372 (−0.602–−0.141) 133	*0.090* *(0.175)*	*0.608*	−0.160 (−0.486–0.166) 62	−0.601 (−0.892–−0.309) 71	*0.440* *(0.247)*	*0.077*
INL mean in % (95% CI) N (eyes)	**1.210** **(0.750–1.670)** 103	**0.326** **(−0.077–0.730)** 133	** *−0.884* ** ** *(0.312)* **	** *0.005* **	0.158 (−0.252–0.569) 61	0.248 (−0.122–0.618) 72	*−0.090* *(0.313)*	*0.775*
ONPL mean in % (95% CI) N (eyes)	−0.078 (−0.359–0.203) 112	−0.317 (−0.578–−0.056) 130	*−0.239* *(1.973)*	*0.226*	−0.312 (−0.640–0.016) 61	−0.308 (−0.613–−0.003) 69	*−0.004* *(0.253)*	*0.986*

Relative annual change rates of retinal layer thickness and volumes were compared among HC and PwMS, respectively, in PPMS and RRMS, after exclusion of extreme lower and upper outliers according to Tukey (1st quartile − 3xIQR < range included < 1st quartile + 3xIQR). Significance was set at *p* < 0.05. *ß*: beta regression; CI: confidence interval; GCIP: ganglion cell and inner plexiform layer; HC: healthy controls; INL: inner nuclear layer; LMU: Ludwig-Maximilians Universität München (Spectralis OCT); MSK: Marianne-Strauß-Klinik (Cirrus OCT); N (eyes): number of eyes included; ONPL: outer nuclear and outer plexiform layer; PPMS: primary-progressive multiple sclerosis; pRNFL: peripapillary retinal nerve fiber layer; PwMS: persons with multiple sclerosis (PPMS and RRMS); RRMS: relapsing–remitting multiple sclerosis; SD: standard deviation; SE: standard error; TMV: total macular volume. ^1^ Generalized linear mixed model using *gender* and *age at OCT_bas_* as covariates to compare HC vs. PwMS. ^2^ Generalized linear mixed model using *gender*, *age at OCT_bas_*, and *disease duration* (initial manifestation to OCT_bas_ in years) to compare MS subtypes.

**Table 4 ijms-24-12872-t004:** Association between ambulation functional score and relative annual change rates of retinal layers.

				RRMS vs. PPMS Stable AS			PPMS Stable vs. Deterioration in AS		
	RRMS Stable AS (*t*-Test)	PPMS Stable AS (*t*-Test)	PPMS Deterioration AS (*t*-Test)	*p*-Value (*t*-Test)	*p*-Value (ANCOVA)	*r*	*p*-Value (*t*-Test)	*p*-Value (ANCOVA)	*r*
Peripapillary Scan (MSK and LMU)									
Relative annual change in pRNFL mean in %, (SD)	−0.766 (2.256)	−1.102 (1.797)	−0.656 (1.019)	0.365	*0.956*	*0.048*	*0.192*	0.362	*0.101*
Macula Scan (LMU)									
Relative annual change in TMV mean in %, (SD)	−0.186 (0.648)	0.001 (0.856)	−0.544 (0.688)	0.239	*0.987*	*0.002*	** *0.025* **	** *0.032* **	** *0.310* **
Relative annual change in GCIP mean in %, (SD)	−0.325 (0.912)	0.278 (1.640)	−1.012 (1.103)	0.060	** *0.004* **	** *0.295* **	** *0.005* **	** *0.006* **	** *0.390* **
Relative annual change in INL mean in % (SD)	0.451 (1.450)	0.724 (2.643)	−0.154 (0.878)	0.516	*0.681*	*0.043*	*0.179*	0.203	*0.187*
Relative annual change in ONPL mean in %, (SD)	−0.291 (1.631)	0.150 (1.672)	−0.979 (1.163)	0.223	*0.856*	*0.019*	** *0.015* **	** *0.020* **	** *0.336* **

The relative annual rates of change in the thickness and volumes of the retinal layers were compared within PwMS and PPMS only, graded by stable vs. deteriorated ambulation functional score over the study period. ANCOVA with covariates of *age at OCT_bas_* and *gender*. No exclusion of outliers. Mean relative annual change rates for the ANCOVA analysis are not shown. *t*-Test stable RRMS vs. PPMS with AS deterioration with *p*-value = 0.008; ANCOVA with covariates of *age at OCT_bas_* and *gender* for stable RRMS vs. PPMS with AS deterioration with *p*-value = 0.181 and *r* = 0.151. Significance was set at *p* < 0.05. AS: ambulation functional score; GCIP: ganglion cell and inner plexiform layer; INL: inner nuclear layer; LMU: Ludwig-Maximilians Universität München (Spectralis OCT); MSK: Marianne-Strauß-Klinik (Cirrus OCT); ONPL: outer nuclear and outer plexiform layer; pRNFL: peripapillary retinal nerve fiber layer; SD: standard deviation; TMV: total macular volume.

**Table 5 ijms-24-12872-t005:** Cox-regression analysis on effects of retinal layer atrophy rates on deterioration in the ambulation functional score and in the EDSS.

	Ambulation Functional Score				EDSS	
	All PwMS		PPMS		PPMS	
	HR (95% CI)	*p*-Value	HR (95% CI)	*p*-Value	HR (95% CI)	*p*-Value
Relative annual atrophy rate of pRNFL (in ‰)	1.009 (0.979–1.040)	*0.550*	1.024 (0.992–1.057)	*0.143*	**1.021 (1.002–1.040)**	** *0.030* **
Relative annual atrophy rate of TMV (in ‰)	**1.265 (1.132–1.414)**	** *<0.001* **	**1.309 (1.155–1.483)**	** *<0.001* **	**1.172 (1.081–1.270)**	** *<0.001* **
Relative annual atrophy rate of GCIP (in ‰)	**1.093 (1.041–1.148)**	** *<0.001* **	**1.112 (1.053–1.174)**	** *<0.001* **	**1.059 (1.016–1.104)**	** *0.007* **
Relative annual atrophy rate of INL (in ‰)	1.017 (0.973–1.062)	*0.456*	1.026 (0.972–1.084)	*0.352*	1.004 (0.974–1.035)	*0.779*
Relative annual atrophy rate of ONPL (in ‰)	1.013 (1.001–1.025)	** *0.036* **	**1.161 (1.086–1.241)**	** *<0.001* **	1.026 (0.977–1.078)	*0.301*

The effect of the relative annual atrophy rates of retinal layers (*relative annual rates of change in retinal layers*x − 1; in ‰) on deterioration in the ambulation functional score (AS) in PwMS (RRMS and PPMS). The EDSS for PPMS subjects was evaluated using a Cox-regression model adjusted for *age at baseline OCT* (OCT_bas_) and *gender*. No exclusion of outliers. Significance was set at *p* < 0.05. CI: confidence interval; EDSS: expanded disability status scale; GCIP: ganglion cell and inner plexiform layer; HR: hazard ratio; INL: inner nuclear layer; ONPL: outer nuclear and outer plexiform layer; pRNFL: peripapillary retinal nerve fiber layer; TMV: total macular volume.

## Data Availability

The data presented in this study are available on reasoned request from the corresponding author if the data exchange is possible with respect to valid data protection laws.

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
