# Peer review of "Inner Retinal Layer Changes Reflect Changes in Ambulation Score in Patients with Primary Progressive Multiple Sclerosis"

_ijms, 2023, doi:10.3390/ijms241612872_

Round 1
Reviewer 1 Report
The authors present an interesting study in which the rate of change in the retinal layer as measured by optical coherence tomography methods is explored as a means of determining disability progression in those diagnosed with multiple sclerosis. Briefly, healthy controls and those with multiple sclerosis were recruited and had their eyes evaluated over a period of an average three years. Within the multiple sclerosis cohort, this evaluation was performed by examining clinical manifestation of multiple sclerosis in having both relapsing-remitting multiple sclerosis and primary progressive multiple sclerosis forms represented. In summary, the authors found that certain measurements of the retina did indeed inform of the progression of the disease, with little to know variation found between either clinical cohort. In short, OCT represents a potentially valuable tool for monitoring multiple sclerosis progression in populations, with it offering several advantages that may reduce the burden on the healthcare system.
In reviewing the manuscript, I made the following observations. The authors should consider the following when preparing a resubmission.
1. While the authors clearly state the numbers in each clinical population, it would be more informative that the number included in each analyses was included in each table to confirm all participants are represented, and not a select number for whatever reason.
2. Can he authors confirm that both eyes, where possible, were evaluated in each participant, or was one representative eye taken for each participant?
3. The supplemental figure for the relative change in INL annually could be improved. The formatting is weak, with the labels incredibly small/illegible, no information on n-number, and no error bars.
Author Response
We thank the reviewer for the positive evaluation of our work and the opportunity to further improve the manuscript. Based on the helpful and thorough evaluations, we have revised the manuscript. In the appendix you will find our detailed point-by-point response.

Reviewer 2 Report
This manuscript by Gernert and colleagues investigated the inner retinal layer changes in ambulation score in patients with PPMS using OCT as a tool. The content of this report is novel. I have some comments that need to be addressed first:
1. The authors may show an OCT picture of retinal layers of patients and health controls, “a picture is worth a thousand words”. It may help the readers without OCT or knowledge structure of retina understand it.
2. I was wondering whether there were any retinal vasculature changes in these MS patients? The authors may give a brief discussion regarding it. I understand the OCTA can detect the vasculature changes while traditional OCT may not.
Thank you!
Author Response

(The authors gave the same response as above.)

Reviewer 3 Report
The present research article by Gernert et al. entitled "Inner retinal layer changes reflect changes in ambulation score in patients with primary-progressive multiple sclerosis" demonstrates the retrospective analysis of a longitudinal OCT dataset of the retinal layers of PwMS and HC from two MS centers in Germany. Authors have systematically evaluated the disability progression in PPMS using OCT and suggested that increasing annual atrophy rates of the inner retinal layers are associated with worsening ambulation. Furthermore, authors claimed that OCT is a robust and side-effect-free imaging tool, making it suitable for routine monitoring of PwMS. There is no specific query on the study and minor english editing is required.
English language is fine.
Author Response
We thank the reviewer for the positive evaluation of our work. There is no specific query on the study.